# Application of CRISPR/Cas9 System for Efficient Gene Editing in Peanut

**DOI:** 10.3390/plants11101361

**Published:** 2022-05-20

**Authors:** Anjanasree K. Neelakandan, David A. Wright, Sy M. Traore, Xingli Ma, Binita Subedi, Suman Veeramasu, Martin H. Spalding, Guohao He

**Affiliations:** 1Department of Genetics, Development and Cell Biology, Iowa State University, Ames, IA 50011, USA; anjanasiva@gmail.com (A.K.N.); wrightd@iastate.edu (D.A.W.); mspaldin@iastate.edu (M.H.S.); 2Department of Agricultural and Environmental Sciences, Tuskegee University, Tuskegee, AL 36088, USA; sytraore@gmail.com (S.M.T.); binitasubedi.bs@gmail.com (B.S.); sveeramasu4712@tuskegee.edu (S.V.); 3College of Agronomy, Henan Agricultural University, Zhengzhou 450002, China; ml25900@126.com

**Keywords:** peanut, CRISPR/Cas9, gene editing, functional genomic tools, hairy root transformation, mutation

## Abstract

Peanuts are an economically important crop cultivated worldwide. However, several limitations restrained its productivity, including biotic/abiotic stresses. CRISPR/Cas9-based gene-editing technology holds a promising approach to developing new crops with improved agronomic and nutritional traits. Its application has been successful in many important crops. However, the application of this technology in peanut research is limited, probably due to the lack of suitable constructs and protocols. In this study, two different constructs were generated to induce insertion/deletion mutations in the targeted gene for a loss of function study. The first construct harbors the regular gRNA scaffold, while the second construct has the extended scaffold plus terminator. The designed gRNA targeting the coding sequence of the *FAD2* genes was cloned into both constructs, and their functionality and efficiency were validated using the hairy root transformation system. Both constructs displayed insertions and deletions as the types of edits. The construct harboring the extended plus gRNA terminator showed a higher editing efficiency than the regular scaffold for monoallelic and biallelic mutations. These two constructs can be used for gene editing in peanuts and could provide tools for improving peanut lines for the benefit of peanut breeders, farmers, and industry.

## 1. Introduction

Peanuts are an economically important crop that is farmed in many regions around the world due to their adaptability to a wide range of climate conditions. Its productivity is severely restrained by several biotic/abiotic stresses worldwide. The early and late leaf spot diseases are fungal diseases that affect peanut productivity, causing yield losses of around 70% in the U.S. and around the world [1]. The powdery mildew disease is another essential fungal disease affecting peanuts in Asia and Africa. Although peanuts are rich in oil, the polyunsaturated fatty acid content is high [2] and has a detrimental effect on human health and the length of shelf storage [3]. Moreover, some proteins in seeds can cause allergenic reactions in humans and have been a public health issue [4]. Therefore, developing functional genomic tools such as CRISPR/Cas9 technology to validate and manipulate important genes associated with disease resistance, abiotic stresses, and nutritional values would provide a platform for the genetic improvement of peanut cultivars. 

CRISPR/Cas9-based gene-editing technology has emerged as a powerful alternative for plant breeding due to its precise, efficient, and targeted gene modification to generate intended traits. With versatile gene-editing tools developed, many potential applications have been performed for crop improvement in a variety of plant species [5,6]. One such application in plant breeding is to modify a gene through loss of function mutation by introducing small indels that result in frame-shift mutations or premature stop codons. The loss of function by disrupting genes is a valuable technique for gene function analysis and trait alteration. For instance, targeted mutagenesis induced by CRISPR/Cas9-based gene editing can turn undesirable genes into desirable ones, such as susceptible genotypes altered to resistance in rice [7], tomato [8], and wheat [9], increase grain weight and protein content in wheat [10] and gain high amylopectin starch content (waxy) in potato [11]. The CRISPR/Cas9 technology has provided a new breeding toolbox to generate genetic variability in plants, making it an excellent complement to conventional breeding. These toolboxes offer a platform for gene editing and the improvement of the crops on which they have been established [6]. However, the application of CRISPR/Cas9 gene-editing technology to peanuts is still limited. 

The cultivated peanut is an allotetraploid stemming from a single hybridization of two diploid ancestors, *Arachis duranensis* (A-genome) and *Arachis ipaensis* (B-genome). The fatty acid desaturase 2 (FAD2) is encoded by two homeologous genes, *AhFAD2A* and *AhFAD2B*, in peanuts. Previously, the well-characterized *FAD2* genes were selected for gene editing in peanuts on the basis of natural mutations in the coding region of *FAD2* genes interrupting the conversion of oleic acid to linoleic acid, leading to a high oleic acid content in peanut seeds that have nutritional and commercial values [12]. The proof-of-concept study showed that the CRISPR/Cas9 technique could generate the same mutations as natural mutations at the hotspot of natural mutations. Although mutations were induced in the coding region, there is still a need to improve the editing efficiency of the CRISPR system, which is suitable for gene editing in peanuts.

The availability of the Cas9 variants and types has broadened the use of the technology, providing other alternatives rather than indels [13]. The fusion of a nucleotide deaminase to the nicked Cas9 (nCas9) allows nucleotide changes at a specific position [14,15,16,17,18], while the fusion of a transcriptional domain, either activator (VP64) or repressor (SRDX), to the nuclease-dead Cas9 (dCas9), allows the up- or down-regulation of gene expression, respectively [19,20]. The type V Cas9 enzyme Cpf1 is a single RNA-guided nuclease that cleaves DNA via a staggered DNA double-stranded break, facilitating robust gene editing and minimizing potential off-target effects in plants [21,22,23]. Various CRISPR/Cas9 constructs have been optimized for their effectiveness in plant species. The gRNA expression cassette is under the *Arabidopsis* U6-26 promoter, while the Cas9 cassette expression is modulated by the *Arabidopsis* ubiquitin promoter [24,25]. The regular sgRNA structure consists of a shortened duplex compared with the native bacterial CRISPR RNA (crRNA) and transactivating crRNA (tracrRNA) duplex and contains a continuous sequence of thymines [26]. The knockout efficiency was significantly increased when the duplex length was extended by 5 bp [26]. In this study, to explore the efficiency of CRISPR/Cas9-based gene editing in peanuts, two constructs (pDW3872 and pDW3877) were generated to induce indels in the *FAD2* gene. The pDW3877 was constructed with the regular gRNA scaffold while the pDW3872 possessed the extended gRNA scaffold. Their functionality and effectiveness were validated in the peanut hairy root transformation system. Targeted mutagenesis induced by CRISPR-based gene editing could demonstrate its utility for basic and applied research for genes of interest in peanuts.

## 2. Materials and Methods

### 2.1. Plant Material

The genotype GT-C20 is a Spanish-type peanut, kindly provided by Dr. Baozhu Guo (USDA/ARS, Tifton, GA, USA). This genotype was used in this study for gene editing due to no natural mutations occurring in *FAD2* genes.

### 2.2. Vector Construction

For the construction of the gRNA cassette, the AtU6-26 promoter from pTF101-AtCa9-GmRCA#1 was used and followed by gRNA regular scaffold and terminator region (pDW3877). To increase the editing efficiency, we also used the optimized gRNA structure instead of the regular scaffold in a separate vector (pDW3872). The authors of [26] demonstrated that the efficiency of mutation was enhanced by extending the duplex of gRNA scaffold by 5 bp combined with mutating the continuous sequence of Ts at position 4 (T > C) after the guide sequence. These two vectors were used to compare the effects of gRNA scaffolds on the editing efficiency in peanuts. For the construction of the Cas9 expression cassette, the coding region of Cas9 was fused in frame with a 3XFLAG at the 3’end and SV40 nuclear localization sequence at both 5′ and 3′ ends. The optimized Cas9 expression cassette was driven by the modified *Arabidopsis* ubiquitin AtUBI promoter and terminated with the CaMV 35s terminator. The right border consists of the pVs1 replicon with the destroyed *Bsa*I site, the ColEI replicon, the *npt*I kanamycin resistance gene, and the spectinomycin resistance gene for bacterial selection. The left border includes the NOS terminator, the *Bar* resistance gene for in planta selection, and the 4× 35s promoter (Figure 1 and Appendix A). The gRNA (5′-ACTTACTCTTCTACATTGCC-3′) was designed using the CRISPR Genome Analysis Tool (CGAT, http://cbc.gdcb.iastate.edu/cgat/, accessed on 27 April 2022) and targeted the position between 179 and 198 bp in the coding region of *AhFAD2* genes. The designed gRNAs were inserted into two vectors at the double *Bsa*I cut sites using the Gibson Assembly. Cloned Cas9-gRNA constructs were sequenced to confirm the insertion of the gRNA. Plasmids were incorporated into *Agrobacterium rhizogenes* strain K599 and used for hairy root transformation.

### 2.3. Hairy Root Transformation

Sixteen sterilized GT-C20 seeds were germinated on ½ MS liquid medium under sterile conditions and grown for approximately one week. The embryo roots and lower hypocotyl were cut from seedlings. Then the remaining upper portion of each seedling was used as explants. The upper portion of hypocotyles from 8 seedlings were infected by *Agrobacterium* with the construct pDW3872 and other 8 hypocotyles infected by *Agrobacterium*/pDW3877 for hairy root transformation following the modified protocol previously described by [12]. Briefly, *A. rhizogenes* was streaked on solid LB Kan_50_ and grew at 28 °C overnight. *A. rhizogenes* cells were scraped from the plate and resuspended in 6 mL of ½ MS liquid. Explants were dipped in the *A. rhizogenes* solutions and incubated for 20 min with occasional inverting. After incubation, explants were transferred to ½ MS media for co-cultivation in the dark at room temperature for 2 days. After co-cultivation, explants were transferred to ½ MS media supplemented with timentin 300 mg/L for *Agrobacterium*-suppressing antibiotics and kanamycin 30 mg/L for selection of transformed roots. Roots were then cultured under fluorescent lights at room temperature with a 16-h photoperiod. After 1.5–2 weeks, transformed roots were harvested from selective media for DNA extraction.

### 2.4. Validation of Mutagenesis

The harvested hairy root sections (3 cm) were collected for DNA extraction. The DNA was extracted from each sample using the CTAB protocol. A PCR analysis was performed to amplify 500 bp amplicon size bearing the gRNA targeted site from both *Ah**FAD*2 genes, the *AhFAD2A* for the A-genome, and the *AhFAD2B* for the B-genome. The amplified PCR products were sequenced by Sanger sequencing at the DNA sequencing facility at Iowa State University. Sequencing results were analyzed, and mutated sequences were analyzed along with the wild-type sequence using the MEGA7 software [27].

### 2.5. Evaluation of the Editing Efficiency

The hairy root system was used to evaluate the editing efficiency of our constructs. Specific primers were designed to PCR-amplify the *AhFAD2A* and the *AhFAD2B*. Each transformed hairy root was considered an independent event. The overall editing efficiency was calculated by dividing the total number of edited hairy roots by the total number of screened hairy roots. The editing efficiency of the monoallelic was determined by dividing the number of the edits in either *AhFAD2A* or *AhFAD2B* by the total number of screened hairy roots, while the editing efficiency of the biallelic was determined using the number of hairy roots, in which both *AhFAD2A* and *AhFAD2B* were edited, in the total number of screened hairy roots.

## 3. Results

### 3.1. Designing and Cloning of the gRNAs into the Constructs

The *AhFAD2A* and *AhFAD2B* sequences were downloaded from PeanutBase (peanutbase.org) and NCBI. Sequence analysis showed that both genes were composed of 1140 bp without an intron sequence and only eleven SNPs between the two genes. The gRNA for this study was designed in the conserved region, avoiding the SNPs targeting both the *AhFAD2A* and the *AhFAD2B* simultaneously. Annealed gRNA products were cloned into the double *Bsa*I sites in the constructs (Appendix A). Sequence analysis of the cloning products confirmed the insertion of the gRNAs in between the AtU6-26 and the regular scaffold in pDW3877 or the extended scaffold in pDW3872.

### 3.2. Types of Edits

Sixty-eight hairy roots were collected from 16 infected hypocotyles, whereas 31 and 37 hairy roots were collected from 8 and 8 transformed hypocotyles by pDW3872 and pDW3877, respectively. The genomic DNA was extracted from each of the transformed hairy roots and non-transformed hairy root samples as controls. The PCR amplification was performed to amplify the amplicon size of 500 bp harboring the protospacer of the *AhFAD2*. The PCR products were sequenced using the Sanger sequencing approach. The analysis of the PCR products showed mutations in the targeted regions of the *FAD2* genes. Both constructs, pDW3877 and pDW3872, induced the same types of edits, including insertion and deletion mutations. The insertion mutations were essentially single to multiple nucleotides on the protospacer, occurring at a lower rate. Notably, a unique insertion of 184 nucleotides was observed in *AhFAD2B* using pDW3877, which was homologous to the root-inducing (Ri) plasmid of the *Agrobacterium rhizogenes*. The deletion mutations were the main type of mutation generated by both constructs. More than 90% of the mutations were deletions, ranging from one nucleotide up to 57 nucleotide deletions on both homoeologous genes (Figure 2A). Chromatogram analysis of sequences clearly showed the presence of mixed peaks after the double-strand breakpoint in the protospacer region due to deletions (Figure 2B). The deleted sequence was shifted from those without nucleotide deletion in the same PCR product, which led to a sequence with mixed peaks in the sequencing chromatogram after the break point. The pDW3872 with the extended scaffold generated more types of deletion than the pDW3877 with the regular scaffold. These results showed that both constructs pDW3877 and pDW3872, can induce indels on the protospacers of the *FAD2* in the peanut genome and can be used for gene editing in peanuts.

### 3.3. Editing Efficiency of Two Constructs

The observed editing efficiency was between 24 and 32% using hairy root transformation. The regular scaffold showed an editing efficiency of 24% while the extended scaffold showed a higher efficiency of 32%. Furthermore, the editing efficiency of individual *AhFAD2A* and *AhFAD2B* was higher with the extended scaffold than with the regular scaffold. Although the same efficiency was observed in *AhFAD2A* and *AhFAD2B* using the regular scaffold, the editing efficiency in *AhFAD2B* was higher than in *AhFAD2A* using the extended scaffold (Table 1). When considering simultaneous mutations in both *AhFAD2A* and *AhFAD2B* genes, the editing efficiency was slightly higher (16%) with the extended scaffold than it was (11.8%) with the regular scaffold. These results indicated that both constructs could induce indels in both *AhFAD2A* and *AhFAD2B* individually or simultaneously with different efficiencies. Although there was no statistically significant difference in the editing efficiency between the two constructs, the construct with an extended scaffold showed relatively high efficiency of mutations compared to the construct with a regular scaffold, suggesting that the extended scaffold would be better used in peanuts. 

## 4. Discussion

The previous test of the CRISPR/Cas9-based gene editing on *FAD2* genes in peanuts was with a designed gRNA targeting the spot of a natural mutation that generated a premature stop codon. Although the same mutations were induced, the deletion mutations were not observed as the generally abundant indel types induced after DNA cleavage and error-prone repair by nonhomologous end-joining (NHEJ) [12]. We aimed to evaluate our modified vectors with different gRNA scaffolds to improve gene-editing efficiency. For this purpose, we chose one target site closed to the 5’ end of the coding region of *FAD2* genes. Two constructs were assembled to contain the Cas9 gene and expression cassettes, with the only difference being their gRNA structures, either extended or regular scaffolds. These two constructs were used to analyze and compare editing efficiency through the hairy root transformation.

The CRISPR/Cas9 expression vectors have been optimized for convenient editing into plants. Several features of these constructs have been replaced to fit in planta expression machinery; therefore, the expression cassettes of the gRNA and Cas9 have been driven within planta expression promoters. The modified *Arabidopsis* U6-26, rice U6-26, and soybean U6-26 promoters, when driving the gRNA cassette, have been shown to be more effective in planta [24], while the Cas9 cassette has been under the expression of the modified *Arabidopsis* ubiquitin, the maize ubiquitin, and rice ubiquitin promoters [25,28,29,30]. In addition, [26] has shown that optimizing the original scaffold by adding five nucleotides is more effective in planta. In both vectors used in this study, the modified *Arabidopsis* U6-26 was driving the gRNA expression cassette while the Cas9 expression cassette was driven by the modified *Arabidopsis* promoter. The pDW3872 harboring the extended gRNA has been shown to be more effective in peanut genome editing. 

The types of edits observed in genome editing in plants are dependent on the plant species used. Studies have shown that different types of mutation occur in various plant species. In rice and *Arabidopsis*, the predominant type of mutation is a 1 bp insertion [31,32]; in tobacco, a few nucleotides’ deletions were the major type of edition [33]. For Chinese kale, short substitutions were the major edits observed [34]. In this study, nucleotide deletions were predominantly the type of edits observed. The observation of the different types of edits in planta, in general, could be explained by many factors, including the mechanism of DNA repair of the species, the selected gRNAs, the constructs, and the delivery methods.

Many systems have been established for the validation of the gRNA/constructs. Among these systems, protoplast transfection has emerged because it provides the possibility of directly delivering the ribonucleoprotein. However, the system is costly and tedious and has the disadvantage of carrying a mixture of edited and non-edited DNA. Nevertheless, the protoplast system has been used to validate the gRNA/construct in *Arabidopsis*, maize, rice, and wheat [35]. The hairy root system has been established in many plant species, including peanut, potato, soybean, and *Medicago* [12,36,37,38,39]. The system is time-consuming and requires tissue culture. However, each developed hairy root is considered an independent event, mutated or not mutated. Moreover, the hairy root transformation provides a stable and more robust validation system. In this study, we used the hairy root transformation to validate the gRNA/constructs.

The *FAD2* gene has a homoeologous gene pair, *AhFAD2A* and *AhFAD2B*, in peanuts. If gene editing could target both homoeologous *AhFAD2A* and *AhFAD2B* simultaneously, it would generate mutants with complete loss of function for the conversion of the oleic acid to linoleic acid and increase the oleic acid content as the evidence observed in natural mutations [40,41,42,43]. In this study, the overall editing efficiency was between 24% and 32% using hairy root transformation for pDW3877 and pDW3872, respectively. Similar editing efficiency was observed in polyploid genome editing, including wheat, at 29–36% [35]. However, the editing efficiency of both genes simultaneously is relatively low, between 12 and 16%. It suggests that more transformations are requested to obtain biallelic edits. Nevertheless, this study showed that the developed constructs could be used to induce indels individually and simultaneously in peanut genomes; therefore, providing tools for functional genomics research and the development of new peanut lines with improved agronomic traits.

## 5. Conclusions

This study developed two constructs with different gRNA scaffolds to induce indels in the genome of peanuts. Results showed that both constructs induced deletions as the major type of mutation at different rates and efficiencies. The editing efficiency of the construct with the extended gRNA scaffold was higher than that with the regular scaffold. Furthermore, these constructs induced mutations in both *FAD2* genes individually and simultaneously. These constructs can be used for basic and applied biology as well. Moreover, the well-characterized homeologous *FAD2* genes were used as a model for studying and optimizing the gene-editing system in the allotetraploid cultivated peanut. Our results confirmed that the CRISPR/Cas9-based gene-editing system would be a promising tool for editing various genes of interest in the peanut genome.

## Figures and Tables

**Figure 1 plants-11-01361-f001:**
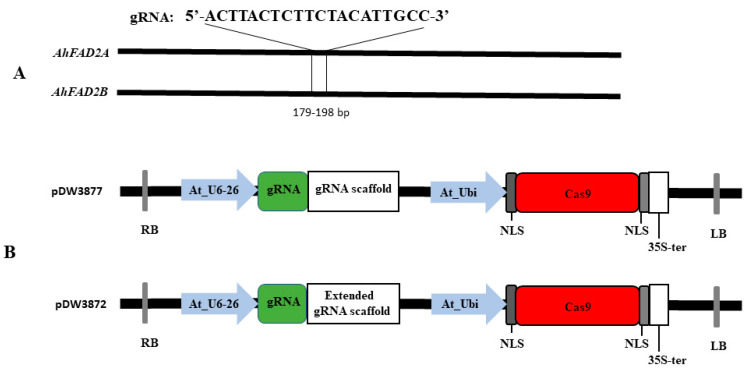
Schematic representation of the constructed vectors. (**A**) The target site and sequence of the designed gRNA in the *AhFAD2A* and *AhFAD2B* genes. (**B**) Representation of the CRISPR/Cas9 binary vector’s structures. The gRNA cassette was driven by the *Arabidopsis* U6-26 promoter, while the Cas9 was controlled by the *Arabidopsis* ubiquitin promoter for both constructs. In the pDW3877, the gRNA cassette was terminated with the regular scaffold, while in the pDW3872, the gRNA cassette was terminated with the extended gRNA terminator.

**Figure 2 plants-11-01361-f002:**
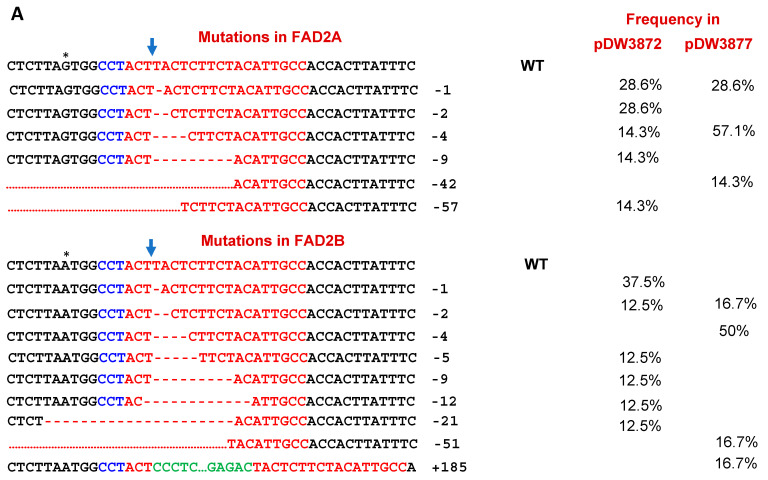
Mutagenesis induced in *FAD2* genes by CRISPR/Cas9 medicated gene editing. (**A**). Two vectors, pDW3872 and pDW3877, induced different mutations with their editing efficiencies. * indicates an SNP between *FAD2A* and *FAD2B*. The target sequences are highlighted in red; insertion sequences in green; PAM in blue. (**B**). An example of chromatograms with 4 bases deletion in *FAD2A* and *FAD2B*. The arrow indicates the break site by nucleases and four bases (in parentheses of wild type sequence) deleted in *FAD2* genes.

**Table 1 plants-11-01361-t001:** Editing efficiency at the coding region of *AhFAD2A* and *AhFAD2B* genes.

Construct	No of Independent Roots Screened	Edited Roots	Editing Efficiency (EE)	EE in FAD2A	EE in FAD2B	Biallelic Efficiency
pDW3872/Cas9-extended scaffold	25	8	32.0%	20.8%	30.4%	16.0%
pDW3877/Cas9-regular scaffold	34	8	23.5%	17.7%	17.7%	11.8%

## Data Availability

Data is contained within the article and Appendix A.

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
