# Peer review of "Application of CRISPR/Cas9 System for Efficient Gene Editing in Peanut"

_plants, 2022, doi:10.3390/plants11101361_

Round 1
Reviewer 1 Report
Dear authors,
The manuscript is fine except a few points:
Please specify the fungal diseases serious to peanut.
On line 68, the fusion to what is discussed?
In line 92, efficiency of transcription is mentioned. How do you think scaffold structure will affect the said efficiency? Please give details or reword.
You claim the extended scaffold is more efficient. Please statistically analyse your data to validate this. Get P values.
Proofread the manuscript. Many singular/plural uses are wrong I believe, among other things.
Author Response
We are grateful for reviewers’ comments that help us a lot to improve our manuscript. Based on their comments, our responses are listed as below.
Reviewer 1
Open Review
Dear authors,
The manuscript is fine except a few points:
Please specify the fungal diseases serious to peanut.
Response: We have mentioned the impact of fungal diseases on peanut production in the section of Introduction.
On line 68, the fusion to what is discussed?
Response: corrected.
In line 92, efficiency of transcription is mentioned. How do you think scaffold structure will affect the said efficiency? Please give details or reword.
Response: corrected using words of “the editing efficiency” instead of words of “efficiency of transcription”.
You claim the extended scaffold is more efficient. Please statistically analyse your data to validate this. Get P values.
Response: We did perform the statistical analysis and found no significant difference in the editing efficiency between two scaffolds. We have explained in the text.
Proofread the manuscript. Many singular/plural uses are wrong I believe, among other things.
Response: grammars were checked and corrected
Reviewer 2 Report
In the manuscript entitled "Application of CRISPR/Cas9 system for efficient gene editing in peanut", the authors briefly presented the results of peanut editing with the use of A. rhizogenes and two types of gRNA scaffolds. In the analyzed hairy roots, editing of the FAD2 gene in the expected region was confirmed by identifying the indel mutation by Sanger sequencing. The authors found differences in the effectiveness of editing depending on the construct used, which in my opinion is debatable in its current form.
The most important comments
Table 1 - no statistical analysis so it cannot be stated whether the effect of the extended scaffold has a significant impact on the editing efficiency. I am afraid that analyzing from 25 to 34 hairy roots it will not be possible to show the differences.
Line 258 - 259 - as before, without statistical analyzes such a conclusion cannot be drawn
Why do the authors refer to the in planta method, which involves infiltration of plant tissues with a bacterial suspension in ex vitro conditions. This may lead to unnecessary doubts.
There is no chapter on the effectiveness of the transformation - how many explants were used, how many hairy roots were obtained, when the effectiveness analysis was performed. Please complete this information as point 3.2
The title does not fit well with the presented studies, which were more about demonstrating greater effectiveness with the use of an extended scaffold
There are reports in the literature on the development of efficient protocols for obtaining transgenic peanut plants, why did the authors decide to perform genetic transformation with A. rhizogenes?
minor remarks:
The introduction should include information from the 280-281 line, so that later there is no doubt about genome-A and genome-B.
The introduction should also be more focus on the influence of differences in gRNA scaffods than on problems caused by stress in peanut, which are not discussed in the research at all
Line 102 and 104 and others - gene names from lower case (also 115 - ubiquitin)
Line 115 Arabidopsis in italics
Line 117 - double dot
Line 76 none “]”
Maybe it is worth including the sequence of both gRNAs to show exactly what was the difference in their scaffold structure (maybe as SM)
Author Response
We are grateful for reviewers’ comments that help us a lot to improve our manuscript. Based on their comments, our responses are listed as below.
Reviewer 2
Open Review
In the manuscript entitled "Application of CRISPR/Cas9 system for efficient gene editing in peanut", the authors briefly presented the results of peanut editing with the use of A. rhizogenes and two types of gRNA scaffolds. In the analyzed hairy roots, editing of the FAD2 gene in the expected region was confirmed by identifying the indel mutation by Sanger sequencing. The authors found differences in the effectiveness of editing depending on the construct used, which in my opinion is debatable in its current form.
The most important comments
Table 1 - no statistical analysis so it cannot be stated whether the effect of the extended scaffold has a significant impact on the editing efficiency. I am afraid that analyzing from 25 to 34 hairy roots it will not be possible to show the differences.
Line 258 - 259 - as before, without statistical analyzes such a conclusion cannot be drawn
Response: We did perform the statistical analysis at the earlier time and found not significant difference in the editing efficiency between two vectors. Both vectors can be used in peanut. Due to the extended scaffold showed a higher efficiency in either FAD2A or FAD2B, it would be better used for peanut.
Why do the authors refer to the in planta method, which involves infiltration of plant tissues with a bacterial suspension in ex vitro conditions. This may lead to unnecessary doubts.
Response: We developed the optimized protocol of hairy root transformation, which is adoptable in soybean and peanut. While the Agrobacterium-mediated transformation in vitro in peanut is time-consuming and genotype-dependent approach.
There is no chapter on the effectiveness of the transformation - how many explants were used, how many hairy roots were obtained, when the effectiveness analysis was performed. Please complete this information as point 3.2
Response: We have added the information on explants and hairy roots used in the sections of Materials and Results.
The title does not fit well with the presented studies, which were more about demonstrating greater effectiveness with the use of an extended scaffold
Response: Previously we published an article about gene editing of FAD2 genes but the editing efficiency was low. The aim of this paper was to explore the application of efficient CRISPR/Cas9-based gene editing in peanut. The title was trying to represent the attempt for efficient gene editing on the same FAD2 genes with new constructed vectors and provide a platform of gene editing on other genes of peanut in the near future.
There are reports in the literature on the development of efficient protocols for obtaining transgenic peanut plants, why did the authors decide to perform genetic transformation with A. rhizogenes?
Response: Most of the existing transformation and regeneration approaches in peanut are genotype- dependent. The Hairy root system has been developed has a robust system for the validation of the gRNA/constructs. We have discussed it in the section of discussion.
minor remarks:
The introduction should include information from the 280-281 line, so that later there is no doubt about genome-A and genome-B.
Response: We added the information in the introduction.
The introduction should also be more focus on the influence of differences in gRNA scaffods than on problems caused by stress in peanut, which are not discussed in the research at all
Response: The reason we described stresses in the introduction is that gene editing would be a good platform of molecular breeding for the genes associated with those stresses in the near future. We also added a description of gRNA structure.
Line 102 and 104 and others - gene names from lower case (also 115 - ubiquitin)
Response: corrected
Line 115 Arabidopsis in italics
Response: corrected
Line 117 - double dot
Response: corrected
Line 76 none “]”
Response: corrected
Maybe it is worth including the sequence of both gRNAs to show exactly what was the difference in their scaffold structure (maybe as SM)
Response: The sequences of both gRNA scaffolds were added to the Supplementary file.